# How to Cope with the Ponseti Method for Clubfoot: The Families’ Standpoint

**DOI:** 10.3390/children9081134

**Published:** 2022-07-29

**Authors:** Daniela Dibello, Giulia Colin, Anna Maria Chiara Galimberti, Lucio Torelli, Valentina Di Carlo

**Affiliations:** 1Unit of Pediatric Orthopedics, Traumatology Giovanni XXIII Children’s Hospital, 70126 Bari, Italy; daniela.dibello@policlinico.ba.it; 2Unit of Orthopaedics and Traumatology, University of Trieste, Strada di Fiume, 34149 Trieste, Italy; giulia.colin@burlo.trieste.it; 3Clinical Department of Pediatrics, Institute for Maternal and Child Health IRCCS Burlo Garofolo, Via dell’Istria 65/1, 34137 Trieste, Italy; annamachigalimberti@gmail.com; 4Clinical Department of Medical, Surgical and Health Sciences University of Trieste, Strada di Fiume, 34149 Trieste, Italy; torelli@units.it; 5Unit of Paediatric Orthopaedics and Traumatology, Institute for Maternal and Child Health-IRCCS Burlo Garofolo, Via dell’Istria, 34137 Trieste, Italy

**Keywords:** clubfoot, families, Ponseti method, questionnaire

## Abstract

(1) Background: The Ponseti Method is the gold standard for the treatment of congenital clubfoot. It is a low-cost treatment consisting in a series of plaster casts, a percutaneous Achilles’ tenotomy and a Mitchell Ponseti brace to wear with a definite protocol. This treatment allows children to be with their families instead of being hospitalized. This advantage is also a challenge for the families that have to follow the protocol at home. This paper aims to analyze the perception, the difficulties and the overcomes of the families during the treatment. (2) Methods: We used a 41 questions questionnaire by Nogueira and Morquende. Questions were answered by families who had already finished the treatment or were still following it. (3) Results: We interviewed 92 families. The worst handling phase appeared to be the cast phase, while the brace seemed more bearable. In total, 57 families overrated tenotomy; (4) Conclusions: Families perceived the Ponseti Method as a quality treatment. The anxiety about the diagnosis played a strong role, but none of the difficulties encountered decreased the treatment outcomes or affected families’ adherence to the protocol. The open-ended answers highlighted that the positive relationship with doctors played a key role in the everyday compliance and the achievement of good results.

## 1. Introduction

In 1948, Doctor Ignacio Ponseti developed a new method for the treatment of congenital clubfoot. The protocol was largely described in his first report in 1963 [1] and in his last report in 1992 [2].

Over the last 30 years, the Ponseti Method became the gold standard for the treatment of clubfoot worldwide [3]. A Cochrane review, including 107 participants, compared it to other non-operative and operative solutions, with the conclusion that the Ponseti method produces better short-term foot alignment and a lower number of relapses [4].

Doctor Ponseti was a promoter of early casting and minimal surgery with the aim of obtaining a well-corrected foot in the shortest time possible. His research led to the understanding that medial column plantarflexion causes the cavus deformity that needs to be corrected at first, with the alignment of the metatarsals. Subsequently, adduction and supination have to be corrected. This first phase consists of gentle manipulations and serial castings, every week. Once cavus, varus, adduction, and supination are corrected, a percutaneous tenotomy is performed to adjust equinus, and then, the last cast is applied. Three weeks after tenotomy, the cast is replaced with an orthosis brace. The Mitchell Ponseti splint (or similar) consists of a pair of shoes with a rotation bar attached to each shoe. The normal foot is positioned in approximately 40° of external rotation and the clubfoot in 60° of external rotation. It has to be worn 23 h/day in the first three months, and then, it can be gradually reduced until it is used only during the night up to 4/5 years of age [5].

The maintenance of the correction is essential to prevent recurrence, which can range from 3.7% [6] to 27.1% [7]. In literature, there is a strong correlation between the recurrence rate and the poor compliance to the brace protocol [8], which parents emphasized to be the most difficult part of the method [9].

This paper aimed to analyze the perception, the difficulties and the overcomes of the families, both during and at the end of the treatment.

The literature has few studies on this topic, and none about an Italian pediatric orthopedic population.

## 2. Materials and Methods

All the included patients were treated at the Institute of Maternal and Child Health, IRCCS Burlo Garofolo of Trieste, in the period between January 2009 and September 2021. The same physicians performed the prenatal consultation, the plaster casts, the percutaneous tenotomy and the follow-up evaluations. The Ponseti protocol was strictly followed. The prenatal diagnosis, if performed during the second trimester [10], was followed by a meeting with the parents, the Gynecologist and the pediatric Orthopaedic with the help of a psychologist. In this occasion, the nature of the deformation, the steps of the treatment and the evidence of the good results of the Method were explained to the parents. In all cases, the treatment started, on average, within the first 10 days of life or at least within the first month [11]. At the first neonatal visit, the severity of the feet deformation has been evaluated with the Pirani Clinical Score [12]. Consecutively, casts were renewed and modified following manipulations every 7/10 days. The percutaneous tenotomy was performed in the operatory room (OR) under light sedation, which has been demonstrated to be completely safe for the baby [13]. At the end of the procedure, 1 cc of Lidocain or Carbocain was locally injected in the proximity of the incision to reduce postoperative pain, and immediately after, the final brace was shaped in the OR in maximum abduction and dorsiflexion of the feet. Three weeks later, when the tendon healed in elongation [14], the brace was replaced at the clinic. At every follow up, all the information about the use of the brace was given to the parents, and physicians were available for telephonic advice or visits, in case of doubts.

For the assessment of parents’ perspective, we used a 41-item questionnaire, validated by Norgueira and Morquende [15]. Parents gave their consent to the anonymous use of their answers.

Questions had yes/no or a multiple-choice answers or were open-ended questions. The questionnaire was divided into topics: description of the patient, prenatal relationship with parents, cast phase, tenotomy, brace phase and results (Appendix A).

For further analysis, we decided to include both children who had finished the treatment and children who were still ongoing to evaluate the presence of differences in parents’ perception of the Ponseti method.

All the children we included in the study had a Pirani Clinical Score of 4 or above. We gave and collected questionnaires in a period of two years during routine follow-up checks.

## 3. Results and Discussion

We offered the questionnaire to 115 families, and we received answers from 92 of them, with an overall response rate of 80%. Among the 92 children treated during the period of evaluation, 76/92 were males (82.6%) and 16/92 females (17.4%), confirming that congenital clubfoot is a male-prevalent condition [16]. Due to the presence, in the study, of children who finished the treatment, as well as children still in therapy, the range of age in our population was between 1 and 10 years of age. In particular, 47 children were less than 5 years of age, and 45 children were 5 or more than 5 years of age. Forty-two (45.6%) patients had just one foot deformed, and 50 (54.4%) had both. Among patients with unilateral clubfoot, the left foot was deformed in 19 patients and the right foot in 23 of them, comparable with the data reported in the literature [17]. The severity of the deformation of every foot has been defined with a Pirani score [12]. In our study group, 76% patients had a Pirani score of 6, while only three patients had a Pirani Score of 4. No children under our care had a Pirani score lower than 4, as this was an exclusion criterion of this study. Among all the patients in the study, we had seven relapses, happening in males aged between 4 and 9 years of age, with a median age of six-years-old when the relapse occurred. We found no correlation between the relapse and families’ social and cultural background or ethnic origin.

### 3.1. Relation with Parents

In total, 78.2% (72 families) of parents knew about their child’s condition before birth, thanks to the ultrasound diagnosis performed during the second trimester. Only 53 (73.6%) of them declared that they had good knowledge or very good knowledge of the Ponseti Method at the first day of treatment, despite the explanations of the pediatric orthopedics or the internet. The other 19 families that had a prenatal diagnosis answered that they did not know much (or knew nothing) about the treatment at its beginning, despite the orthopedic or pediatric counselling. We explained this data knowing that prenatal diagnosis was a very shocking and delicate moment, as it was the first time parents faced the fact that their child had a problem. We tried to give all the information about the condition and the treatment, indeed, and besides the counselling in the prenatal period, we were always available to answer every doubt. We asked ourselves if the parents who knew about their children’s condition before birth were more inclined to strictly follow the Ponseti method, as they had more time to accept the diagnosis, in order to lower the occurrence of relapses. We found no statistical correlation between prenatal diagnosis and relapses: in our data, 5 out of 7 children who relapsed had a prenatal diagnosis of clubfoot, and this was independent to the reason why parents were inclined to adhere to the protocol or not. The whole group interviewed reported the careful and exact adherence to the brace protocol. Probably, this and the close follow-up checks helped to generate the poor percentage of relapses we evidenced in our population of study [8,9].

### 3.2. Cast Phase

A total of 37 families (40.2%) affirmed that their children had some problems adapting to the first cast, but all of them assured that they were well-prepared for that event. This answer confirmed that the continuous dialogue between families and doctors plays a key role in maintaining a good compliance and helps parents and children to have a positive attitude when difficulties during the treatment approach arise. The majority of the families (91%) had great support and help from their close relatives and friends; despite that, everyday activity, such hygiene and bathing habits, were found difficult or very difficult to pursue by 38% of them. Moms who were able to breastfeed the baby with no complications included 69.2% of the sample. Only 37% (34/92) of children had minor skin problems, such as sore and erythema, which had promptly been treated during the weekly cast change and had no long-term complications.

### 3.3. Tenotomy

The percutaneous Achilles tenotomy is a very simple and quick procedure and was performed in 94.9% of the studied patients. Our percentage is a little higher than what reported in literature, where the percentage of tenotomy is reported to be between 80% and 90% [18,19]. Only five children did not need the correction of an equinism with tenotomy. A total of 73 out of 87 families said that they were informed about the procedure, and 57 of them declared that it was simpler than anticipated. As a surgical procedure, the families could misperceive the procedure and feel it as a moment of anxiety. After surgery, when the parents realize that the child did not suffer during and after surgery, they tend to reconsider the whole procedure.

### 3.4. Brace

Almost the entire group of families (98.7%) affirmed they understood the importance of using the brace. Forty-three (46.7%) families declared they had difficulties during the first week of the use of the brace, and, in general, the most challenging period appeared to be during the first six months of its use (38/92 families). However, we were not able to conclude if the perception had changed during the years because of the sample size of the age classes, which is not comparable.

Long-term difficulties using the brace were reported to be skin problems (29 patients) and finding a comfortable position to sleep (4.6%, 41 children); nevertheless, 77% of parents did not have problems in leaving their house when the children wore the brace, and the babies were able to kick (82%) and stand (71.8%) while wearing the brace. (Table 1)

A total of 64% of families considered the cast phase more difficult than the brace phase (four parents did not answer to the question). If we divided our group by the age criterion, in the group with 47 children of less than 5 years of age, 29 (61.7%) declared the cast phase to be the worst. In addition, also the families of children that finished the treatment said that the cast phase was worse than the brace phase, with a percentage of 65.8%. The reason can be found in the difficulties during hygiene routine and in the fact that there is no free time from the cast during the cast phase. Moreover, during this phase, parents have an active role, requesting to accurately monitor the peripheral vascular status of the foot that could be compromised if the cast is built too tight on the leg. For all these reasons, the cast phase can be felt as a very anxious time for the parents. During the brace phase, the parents can remove the brace for complete hygiene of the children, and they can gradually dismiss the brace. Moreover, families start to see a better shape of the foot and can finally recognize the results of the method and feel all the efforts as worth it, encouraging them to continue in the same way. This consideration differs from the evidence of Ramirez et al. [9], which found that the brace phase was the most challenging and that the poor adherence was the main cause of recurrences [20]. In accordance with this study, another one identified that the parents need educational support to understand all the phases of the process of treatment. The families of the cited study exposed the effects on daily living and accommodations made to follow the treatment. Parents perceived that the role of the healthcare provider was to present how-to information, specify consequences of the risks, promote awareness, provide encouragement and support and remind them of positive effects to be expected when treatment is complete [21]. This may be a tip for doctors to use for the brace phase and for the cast phase as well, since a strong physician–family partnership is an important factor for the adherence to bracing [22]. Among the children who finished the treatment (45 children > 5-years-old), all of them practiced sports such as karate, basketball, soccer, ballet, swimming and skating, and no one haf any physical limitation if compared with other children of the same age.

All the families declared their satisfaction with the results of the treatment, except two that did not answer the question, and all of them would recommend the treatment to other families.

## 4. Conclusions

In the perception of the interviewed families, the Ponseti Method is an effective treatment. The anxiety about the diagnosis and the fear of the results of treatment play a strong role, mostly in the first period of treatment. The problems that every family encountered are related to hygiene during the cast phase, which was felt to be the worst part of the method. The brace phase was more bearable, in spite of the little skin problems. None of the difficulties encountered by the families decreased the treatment outcomes, because they did not affect the families’ adherence to the protocol, which continued as well. We also believe that the support families, found in the confrontation among others who had the same treatment, have been a cornerstone of help in the difficult periods. The open-ended answers highlighted the positive and helpful relationship between doctors and patients, we believe that is the key in the everyday compliance of the families to treatment and helps achieve the positive long-term results.

## Figures and Tables

**Table 1 children-09-01134-t001:** Difficulties with the use of the brace.

Skin Problems	Sleep Problems	Transportation Problems	Leg Motility Problems
14.4%	44.6%	23%	18%

## Data Availability

Not applicable.

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
