# Peer review of "How to Cope with the Ponseti Method for Clubfoot: The Families’ Standpoint"

_children, 2022, doi:10.3390/children9081134_

Round 1

Reviewer 1 Report

Thank you for your hard efforts and work for this paper.

Dibello et al have presented a paper on the ponseti method of treating patients with clubbfoot and particularly "How to cope with the ponseti method for clubfoot: the families standpoint."

I believe this answers an integral question as paediatric orthopaedics closely intertwines the patient and parent care.

Overall the paper is easy to follow and answers the aim.

There are minor grammatical and awkward English phrasings throughout the paper that will require help from a native English speaker to ensure an easy read. Additionally, I would recommend splitting up the results and discussion sections. 

Introduction: clear and provides background. However, there is no information as to why this question needs to be asked and answered. There must be explanation of the deficiency in the literature and if others have asked the question, what this paper brings to the literature.

line 14, 35- gold standard does not have capital letters 

line 53- are the commas supposed to be decimal points? (seems to be a systemic issue)

line 54 - no capital for methods. There have been multiple minor grammatical issues until now. It may be beneficial to have a native english speaker quickly read and correct minor changes for an easier read. Thank you

Materials and methods: well done but see below

line 105- inclusion and exclusion criterion should be in the methods first

Results and discussion: I would recommend splitting up the two sections for more clarity 

Line 109- no need for "the" in "the 78.2%" - also applies for consequent use in the paper 

Author Response

Thank you for you time and suggestions.

We modified our paper thank to your tips and it sure enriched our work.

Thank you

Reviewer 2 Report

This manuscript is survey based study the elicits a patients' family perspective on coping with the Ponseti Method used for clubfoot correction. It is an agile read and adds to clubfoot literature. Overall, in its present form, it is moderately written and can benefit from professional English-language editing, as at times grammatical errors hinders its’ flow. Following are my suggestions that I hope the authors take on board:

Introduction:

The authors should provide some background on the challenges reported by families that are published in the literature and how this study is different (https://www.ncbi.nlm.nih.gov/pmc/articles/PMC3215110/), or if no reports are published, how their study add to this deficiency.

Methods:

Methods section could benefit from sub-heading (participants, inclusion/exclusion criteria, intervention, measure such as patient/Pirani questionnaire, and statistical analysis) and further details (such as who did the casting; residents, fellows, attending, techs etc.; surgery details, patients' family education/income level) will benefit our readers.

Is there any relationship between family coping survey and change in Pirani score or length of treatment? i.e. does poor pirani score or increased length of treatment relate to poor coping?

Ethical Concerns: The study IRB states year 2011 but inclusion date was from year 2009? Please explain why this discrepancy. Were patients included in study prior to IRB approval?

Results:

Discussion needs to be separate. Please provide a demographics table.

Line 99: Graphic 1 is missing in the manuscript.

Discussion:

Summarize your findings. Detail why patients' family perspective is important and how the finds from this study are unique.

Provide strength/limitations of this study.

Line 174: Ramirez et al. deals with adherence and does not report patients' family coping.

Discuss more studies addressing this issue. (see citation above in introduction. check its references or search pubmed - https://pubmed.ncbi.nlm.nih.gov/?term=%22Clubfoot%2Ftherapy%22%5BMesh%5D+AND+%22family%22)

Conclusion:

What is the take home message and any future studies that are needed in this field.

Author Response

Thank you for your time an your suggestion.

We wanted to explain that the study desing has been made between 2010 and 2011 and approved in 2011. Then we decided to retrospectively include patient that started the treatment from 2009.

We used you suggestion to improve our paper and for that we thank you.

Round 2

Reviewer 2 Report

The authors have addressed my ethical questions.